# Collaborating Heterogeneous Natural Language Processing Tasks via Federated Learning

## Abstract

The increasing privacy concerns regarding personal private data promote the development of federated learning (FL) in recent years. However, the existing studies on applying FL in Natural Language Processing (NLP) are not suitable for coordinating participants with heterogeneous or private learning objectives. In this study, we further broaden the application scope of FL in NLP by proposing an Assign-Then-Contrast (ATC) framework, which enables clients with heterogeneous NLP tasks to construct a FL course and learn useful knowledge from each other. Specifically, clients are suggested to first perform local training with the unified tasks assigned by the server rather than using their own learning objectives, which is called the Assign training stage. After that, in the Contrast training stage, clients train with different local learning objectives and exchange knowledge with other clients who contribute consistent and useful model updates. We conduct extensive experiments on six widely-used datasets covering both Natural Language Understanding and Natural Language Generation tasks, showing that ATC framework achieves significant improvements compared to several baseline methods. We will release the source code for promoting further research.

## 1 Introduction

Learning from a large quantity of data is one of the critical factors for the great success of large machine learning models (Devlin et al., 2019; Lewis et al., 2020; Raffel et al., 2020; Brown et al., 2020) in a wide range of Natural Language Processing (NLP) applications. However, with the increasing privacy concerns among the public and the implementation of data protection regulations (e.g., General Data Protection Regulation[1]), data owners are required to collect and store personal data in a way that ensures privacy while training machine learning models.

Motivated by such privacy protection requirements, federated learning (FL) (McMahan et al., 2017; Yang et al., 2019) has been proposed to collaboratively train models from decentralized data in a privacy-preserving manner, which has gained rapid popularity in both academia and industry. Previous studies (Hard et al., 2018; Ge et al., 2020; Qin et al., 2021; Passban et al., 2022) on the adoption of federated learning for NLP applications mainly follow the framework suggested by FedAvg (McMahan et al., 2017): Towards the same learning objective, clients independently train the model based on local data and send their model updates to a server for federated aggregation.

Adopting such a FL framework brings several limitations in real-world NLP applications. Firstly, it is typically expected that only participants with the same learning objective can engage in a FL course for jointly training models. However, there are numerous practical use cases where participants with heterogeneous tasks can greatly benefit. For example, multiple companies may gather text data from similar domains, such as news, but utilize these data for different downstream tasks, such as event discovery and sentiment analysis. Secondly, given that a consensus on the learning objectives should be achieved among participants beforehand within such a framework, it might not be suitable for participants who want to keep their learning objective private due to privacy concerns or conflicts of interest. For example, several investment institutes and quantitative trading companies aim to collaboratively learn knowledge without sharing their objectives (e.g., event discovering,

---

[1] https://gdpr-info.eu

factors mining, etc.) since it might hurt the interests of their institutes and companies. These limitations severely impede the further advancement of federated learning in NLP applications, as the goal of federated learning is to bridge isolated data islands rather than just coordinating participants with the same and known learning objectives.

To address these limitations, in this paper, we propose a novel FL framework for NLP applications, named ASSIGN-THEN-CONTRAST (denoted as ATC), which enables participants with heterogeneous or private learning objectives to learn from shared knowledge via federated learning. To be more specific, the proposed framework suggests a two-stage training paradigm in the constructed FL courses, including: (i) ASSIGN. In this stage, the server assigns unified tasks (e.g., self-supervised learning objectives such as masked language modeling and denoising reconstruction) of local training to clients, in addition to broadcasting up-to-date global models. This allows clients to perform local training with the assigned tasks, learning knowledge from local data without using their own learning objectives. (ii) CONTRAST. Clients perform local training according to their own learning objectives and simultaneously optimize a contrastive learning objective to exchange useful knowledge with each other. The server strategically aggregates model updates based on the measured distances among clients to make better use of these model updates. Note that these two training stages complement each other. In the ASSIGN training stage, clients perform local training with the assigned self-supervised objectives, and their model updates are more related to their data domains; While in the CONTRAST training stage, clients train with their own objectives, thus their model updates are more specific to their downstream tasks.

We conduct empirical evaluations on six widely-used datasets, including various Natural Language Understanding (NLU) and Natural Language Generation (NLG) tasks such as text classification, question answering, abstractive text summarization, and question generation. The experimental results demonstrate the effectiveness of the proposed framework ATC in helping clients with heterogeneous or private learning objectives to participate and benefit from a FL course. Compared to several baseline methods, constructing FL courses with ATC achieves noticeable improvements for clients with heterogeneous learning objectives.

## 2 PRELIMINARY

Federated learning (FL) (McMahan et al., 2017; Yang et al., 2019) is a training paradigm that enables multiple participants to collaboratively train global models without directly sharing their local data. Formally, given that there exists one server and $N$ clients in a FL course, each client $n$ locally stores the private data $D_n$ with an amount of $|D_n|$, which won't be shared due to privacy concerns. Here we focus on horizontal federated learning where the feature spaces of clients' local data have been aligned. The following objective function is minimized by the participants:

$$F(\boldsymbol{w}) = \sum_{n=1}^{N} \frac{|D_n|}{\sum_{i=1}^{N} |D_i|} F_n(\boldsymbol{w}), \tag{1}$$

where $\boldsymbol{w}$ denotes the parameters of global model, and $F_n$ denotes the objective function of client $n$.

When adopting FL in NLP applications, most previous studies (Ge et al., 2020; Qin et al., 2021; Passban et al., 2022) assume that clients' learning objectives are the same. For example, Passban et al. (2022) adopts federated learning to train mixed-domain models conducting the same machine translation task. However, in practical NLP applications, participants involved in a FL course might have heterogeneous or private learning objectives, including various NLU and NLG tasks.

In this study, we propose to allow participants with heterogeneous or private learning objectives to learn from shared knowledge via federated learning.

## 3 METHODOLOGY

In this section, we introduce the proposed ASSIGN-THEN-CONTRAST (ATC) framework, which consists of two training stages called ASSIGN and CONTRAST. An overview of the proposed ATC framework is illustrated in Figure 1. First of all, in Section §3.1, we present the preparation of model backbones to enable clients with heterogeneous learning objectives to participate in a FL

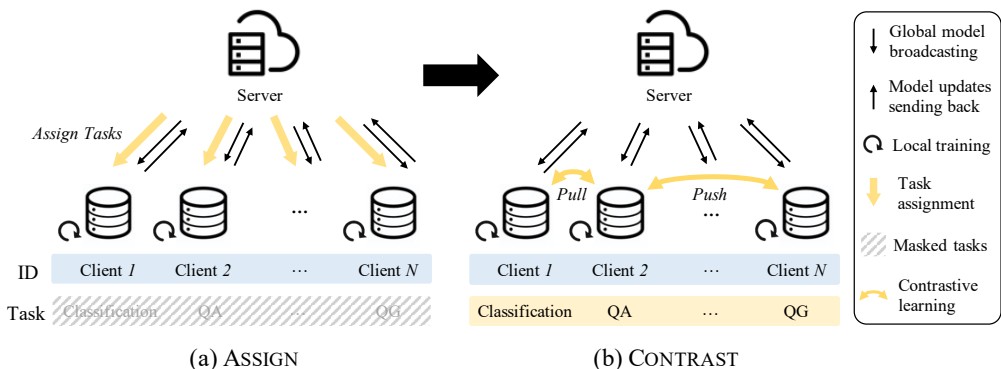

Figure 1: Overview of the proposed framework ATC, which consists of (a) the ASSIGN training stage: clients perform local training with the assigned tasks; and (b) the CONTRAST training stage: clients exchange useful knowledge with each other via optimizing a contrastive loss.

course. Then in Section §3.2, we describe the ASSIGN training stage, in which the server organizes the joint learning via assigning unified tasks to clients at each training round. After that, in the CONTRAST training stage, clients learn useful knowledge from others with the help of a contrastive learning objective, as described in Section §3.3.

## 3.1 MODEL BACKBONES

To enable clients with heterogeneous learning objectives to participate in a FL course, including both NLU and NLG tasks in the field of NLP, the model backbones shared among clients are required to be aligned. Revisiting the Transformer-based (Vaswani et al., 2017) architectures and the standard Seq2Seq framework, the widely-used model backbone contains two fundamental components, i.e., the encoder and the decoder. The encoder model can be applied to NLU tasks, while the entire encoder-decoder model can be applied to NLG tasks.

Based on such insights, we adopt the encoder-decoder architecture as the model backbone in the proposed ATC framework. For clients that only maintain the encoder for NLU tasks, the model backbone can be easily extended by the existing techniques such as BERT2BERT (Rothe et al., 2020), which initializes both the encoder and decoder with the weights of BERT (Devlin et al., 2019).

In this way, the clients' model backbones that are federally learned can be aligned. Additionally, clients can maintain some private layers for personalization, such as hidden layers and classifiers, as illustrated in Figure 2.

## 3.2 ASSIGN: TRAINING WITH ASSIGNED TASKS

The ASSIGN training stage aims to enhance the exchange and integration of the general knowledge contained in clients' local data. However, the diversity and inaccessibility of clients' learning objectives can hinder this progress. To address this, we design a local training approach that is agnostic to clients' learning objectives: clients locally update the received global models based on their local data and the tasks assigned by the server, as shown in the left subfigure of Figure 1.

To be more specific, at each training round of FL, the server broadcasts the up-to-date global model and one of the prepared tasks to clients. The prepared tasks are expected to be beneficial for clients with heterogeneous tasks and not cause additional privacy leakage issues. Most of the supervised learning tasks are unsuitable unless the server can annotate clients' local data without accessing them. All these requirements inspire us to adopt the pre-training tasks in NLP, which are general and beneficial for various NLP downstream tasks, and serve in a self-supervised manner that clients can give annotations by themselves.

In the proposed framework ATC, two widely-used NLP pre-training tasks, including Masked Language Modeling (MLM) (Devlin et al., 2019) and Denoising Reconstruction (DR) (Lewis et al., 2020), are adopted as examples. Note that ATC allows for a variety of tasks to be used in the AS-

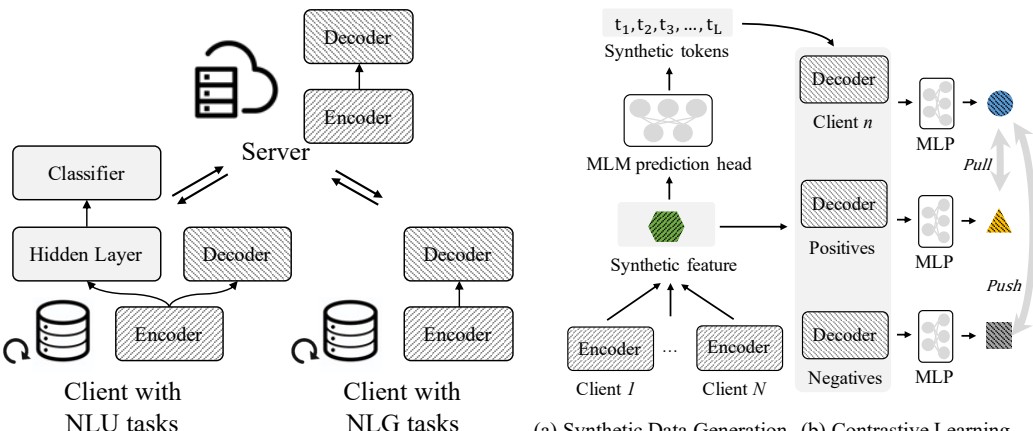

Figure 2: The adopted model backbones of clients with NLU or NLG tasks in the proposed ATC framework.

Figure 3: In the CONTRAST training stage, (a) synthetic dataset generation and (b) contrastive learning.

SIGN training stage. During each training round, the server aggregates the trainable parameters of the global model based on the assigned task. For example, when the server assigns MLM to clients, the parameters of the encoder would be updated and aggregated; And when the assigned task is DR, the updated and aggregated parameters would include both the encoder and decoder.

With ATC framework, participants can construct a FL course without revealing or aligning their own learning objectives. Participants learn from shared knowledge through federal training with assigned tasks, which can also mitigate the heterogeneity in different learning objectives.

**Extension with Clustering Algorithms**   Although the ASSIGN training stage tackles the heterogeneity in clients' learning objectives to some extent, the heterogeneity brought by the non-IID distributions among clients' data might lead to sub-optimal solutions of the learned global model. Such a problem has been studied in personalized federated learning recently (Li et al., 2021a;b; Sattler et al., 2021; Xie et al., 2022), which motivates us to adopt clustering algorithms for alleviating the gradient conflicts when performing federated aggregation for to further improvement.

Specifically, we adopt the agglomerative clustering algorithm (Müllner, 2011) to group clients with similar data domains in a hierarchical manner, which is based on the cosine similarity of model updates. Before federated aggregation, the server clusters clients into several groups based on their model updates. Only the model updates from clients in the same cluster are aggregated. Formally, the server maintains the personalized model for each client $n$ at $t$-th training round as follows:

$$\boldsymbol{w}_n^{(t+1)} \leftarrow \boldsymbol{w}_n^{(t)} + \sum_{i \in \mathcal{C}_n} \frac{|D_i|}{|D_{\mathcal{C}_n}|} \Delta \boldsymbol{w}_i^{(t)}, \tag{2}$$

where $|D_{\mathcal{C}_n}| = \sum_{j \in \mathcal{C}_n} |D_j|$ with $\mathcal{C}_n$ representing the cluster that client $n$ belongs to, and $\Delta \boldsymbol{w}$ denotes the model updates, i.e., $\Delta \boldsymbol{w}_n^{(t)} = \boldsymbol{w}_n^{(t)'} - \boldsymbol{w}_n^{(t)}$ ($\boldsymbol{w}_n^{(t)'}$ implies the model after local training).

It's worth noting that rich types of personalized federated learning algorithms can be used in the ASSIGN training stage, and here we just introduce a representative one and focus on the proposed framework. Further, we provide empirical results and analysis in Section §4.4 to show the effectiveness and contribution of the adopted clustering algorithms in improving the information exchange among similar data domains.

### 3.3   CONTRAST: SHARING KNOWLEDGE VIA CONTRASTIVE LEARNING

After the ASSIGN training stage, clients continue to perform local training with their own learning objectives and adopt a contrastive learning technique to exchange useful knowledge with each other in the CONTRAST training stage.

Applying contrastive learning in the context of FL is more challenging than centralized training (Radford et al., 2021; Cao & Wang, 2021), since it is necessary to prepare or generate datasets for clients to calculate the contrastive loss without causing privacy leakage (Tan et al., 2022b).

One straightforward solution is to use an extra public dataset, which can be fed into clients' models to generate the aligned representations or logits for computing the contrastive losses without raising new privacy leakage issues. However, considering the non-IID distributions among clients' local data in practical NLP applications, it is intractable to find such an eligible public dataset, as pointed out by previous studies (Tan et al., 2022a).

To handle the scenarios where an eligible public dataset might not available, we propose to generate a synthetic dataset in the CONTRAST training stage, as shown in Figure 3. Specifically, each client randomly chooses one instance from its local data, feeds it into its local model, and takes the output of the encoder's last layer. These outputs (vectors in the hidden space) are sent to the server and mixed up to generate a synthetic feature, inspired by the main idea in Zhang et al. (2017). The synthetic feature is passed through an MLM prediction head (here we can reuse the MLM prediction head in ASSIGN training stage or load from a pre-trained language model (Devlin et al., 2019)) to reconstruct the corresponding tokens of this synthetic instance. This process is repeated to generate a synthetic dataset with sufficient quantities. After that, the server broadcasts the synthetic dataset to all the participants.

Furthermore, inspired by previous studies (Cao & Wang, 2021), each client connects a multi-layer perceptron (MLP) to the decoder's last layer. At each FL training round, clients feed the synthetic dataset into their decoders to get the averaged outputs of the MLP as the summarized representations, denoted as $\boldsymbol{h}$. Thus, the contrastive learning loss (Chen et al., 2020) can be calculated as:

$$\mathcal{L}_n = -\log(\frac{s_n^+}{s_n^+ + s_n^-}),$$
$$s_n^+ = \exp(\text{sim}(\boldsymbol{h}_n, \frac{1}{|\mathcal{C}_n^+|}\sum_{i\in\mathcal{C}_n^+}\boldsymbol{h}_i)/\tau), \quad s_n^- = \sum_{j\in\mathcal{C}_n^-}\exp(\text{sim}(\boldsymbol{h}_n, \boldsymbol{h}_j)/\tau), \tag{3}$$

where $\boldsymbol{h}_n$ denotes the summarized representations provided by client $n$, $\text{sim}(\cdot, \cdot)$ denotes the cosine similarity function, and $\tau$ denotes the temperature hyperparameter. The set $\mathcal{C}_n^+$ and $\mathcal{C}_n^-$ represent the positives and negatives for client $n$ in contrastive learning, respectively. The approach to identifying positives and negatives can vary, and here the distances among clients are measured by the cosine similarity of their model updates, and then one's $K$ ($K$ is a hyperparameter) closest neighbors are regarded as positives while others as negatives. Note that the contrastive learning loss is applied in conjunction with the task-specific objective at each client.

In the CONTRAST training stage, users can flexibly balance model utility and privacy protection strength. Based on fundamental privacy protection (no sharing of data directly) provided by ATC, users can choose to adopt some privacy protection mechanisms, such as differential privacy, on the summarized representations $\boldsymbol{h}$ before sharing. Intuitively, sharing hidden representations is safer (He et al., 2020; Gong et al., 2021) and more robust against privacy attacks (Shao et al., 2023) than sharing the entire model or gradients. Besides, the proposed framework can be enhanced by various approaches, such as norm clipping (Sun et al., 2019) and robust aggregation rules (Cao et al., 2019), to mitigate backdoor attacks.

Finally, in the federated aggregation, each client is aggregated with its K closet neighbors based on the gradient similarity, which can be formulated as below:

$$\boldsymbol{w}_n^{(t+1)} \leftarrow \boldsymbol{w}_n^{(t)} + \sum_{i\in\mathcal{C}_n^+}\frac{|D_i|}{|D_{\mathcal{C}_n^+}|}\Delta\tilde{\boldsymbol{w}}_i^{(t)}, \tag{4}$$

where $|D_{\mathcal{C}_n^+}| = \sum_{j\in\mathcal{C}_n^+}|D_j|$, and $\tilde{\boldsymbol{w}}_n^{(t)}$ denotes the model updates when client $n$ performs local training to minimize both its learning objective and the contrastive loss defined in equation 3. Notably, to prevent the potential leakage of local learning objectives, each client only sends the model updates of the encoder to the server in the CONTRAST stage.

## 4 EXPERIMENTS

### 4.1 DATASETS AND METRICS

We adopt six widely-used datasets for conducting a series of experiments. These adopted datasets contain various representative NLU and NLG tasks, such as text classification, question answering, text summarization, and question generation.

Specifically, we adopt two text classification datasets, i.e., IMDB (Maas et al., 2011) and AG-News (Zhang et al., 2015), which are collected from domains of review and news, respectively. For question answering, we adopt the SQuAD (Rajpurkar et al., 2016) and NewsQA (Trischler et al., 2017) datasets, which come from Wikipedia and news, respectively. Besides, we adopt a text summarization dataset CNN/DM (Hermann et al., 2015) and a question generation dataset MSQG (Liu et al., 2021), which both belong to the news domain. Each dataset is randomly partitioned into several clients according to its quantities, as summarized in Table 1.

Table 1: Statistics of the adopted datasets.

| Dataset | Task | Domain | Num. of Clients | Train/Validation/Test |
|---------|------|--------|-----------------|----------------------|
| IMDB | Text Classification | Review | 1 | 22,500/2,500/25,000 |
| AGNews | Text Classification | News | 3 | 108,000/12,000/7,600 |
| SQuAD | Question Answering | Wikipedia | 3 | 117,286/13,033/11,873 |
| NewsQA | Question Answering | News | 2 | 66,744/7,416/4,212 |
| CNN/DM | Text Summarization | News | 5 | 258,398/28,715/13,368 |
| MSQG | Question Generation | News | 4 | 178,250/19,808/11,008 |

For evaluation metrics, we use accuracy (ACC) in text classification tasks, exact match (EM) and F1 score in question answering, and ROUGE-1/2/L (R1/2/L) (Lin, 2004), BLEU-4 (B4) (Papineni et al., 2002) and METEOR (MET) (Banerjee & Lavie, 2005) in text summarization and question generation. All these clients with heterogeneous NLP tasks participate in one FL course in the experiments. The reported results of each dataset are the average scores of all the belonging clients.

### 4.2 BASELINES

We compare the proposed ATC with the three categories of baselines, including:

- *Local Training*, denoted as **ISOLATED**, where clients independently train local models without exchanging any information with each other.

- *Vanilla FL and its variants*, including: (i) **FEDAVG** (McMahan et al., 2017), which proposes each client locally trains the received global model based on its own data and learning objective, and sends the model updates to the server for federated aggregation. Here federated aggregation can be only performed on the sub-model that is consistently maintained by all clients; (ii) **FEDAVG-ST**, which constructs multiple FL courses with FEDAVG independently. Each FL course only involves clients with the same or similar (i.e., text summarization and question generation) tasks; (iii) **FEDPROX** (Li et al., 2020), which adds a proximal term to each client's loss function to reduce the instability caused by data heterogeneity in FL.

- *Personalized FL*, including: (i) **DITTO** (Li et al., 2021a), which trains local and global models simultaneously and fuses the local model update with the global model; (ii) **FEDBN** (Li et al., 2021b), which suggests not sharing the batch/layer normalization parameters with others to address the non-IIDness among clients' local data; (iii) **PERCFL** (Sattler et al., 2021), which proposes personalized clustered FL method based on bi-partitioned clustering; (iv) **SPFL** (Xie et al., 2022), which defines the relationships among clients based on the similarity of their contributed gradients.

### 4.3 IMPLEMENTATION DETAILS

We use the weights of uncased $\text{BERT}_{\text{TINY}}$ (Turc et al., 2019) to initialize the encoder and decoder of the BERT2BERT model, which is provided by Huggingface (Wolf et al., 2020). For all the conducted

Table 2: The comparisons between the proposed ATC framework and baselines. **Bold** and underlined indicate methods with the best and second-best performances, respectively.

| Method | IMDB (ACC) | AGNews (ACC) | SQuAD (EM/F1) | NewsQA (EM/F1) | CNN/DM (R1/R2/RL) | MSQG (RL/B4/MET) | Average |
|---|---|---|---|---|---|---|---|
| Isolated | 78.81 | 92.04 | 43.25/45.14 | 12.01/22.39 | 37.01/15.14/33.33 | 21.32/1.61/11.77 | 45.38 |
| FedAvg (McMahan et al., 2017) | 79.14 | 92.20 | 46.17/48.88 | 15.65/24.54 | 32.64/11.35/29.59 | 21.98/1.67/12.93 | 45.95 |
| FedAvg-ST | 79.38 | 92.75 | 45.28/48.79 | 18.68/28.47 | 35.53/13.78/32.09 | 23.92/1.96/13.91 | 47.19 |
| FedProx (Li et al., 2020) | **79.88** | 92.47 | 46.28/49.13 | 14.62/23.57 | 27.82/7.89/25.45 | 19.08/0.96/10.89 | 44.97 |
| Ditto (Li et al., 2021a) | 79.48 | 92.78 | 42.83/46.49 | 18.66/30.03 | 35.41/13.77/32.02 | 22.99/1.79/13.31 | 46.84 |
| FedBN (Li et al., 2021b) | 79.66 | 92.58 | 45.32/48.38 | 16.64/26.10 | 32.82/11.54/29.72 | 22.10/1.65/12.96 | 46.23 |
| PerCFL (Sattler et al., 2021) | 77.72 | 92.32 | 41.76/46.60 | 21.84/**34.43** | 37.21/15.28/33.51 | 25.96/2.45/15.14 | 47.59 |
| SPFL (Xie et al., 2022) | 77.39 | 91.91 | 39.96/44.34 | 20.11/32.21 | 36.86/15.00/33.21 | 25.23/2.30/14.68 | 46.67 |
| ATC (ours) | 79.72 | 92.86 | 46.35/49.83 | **22.58**/34.24 | **37.88/15.79/34.13** | **28.14/3.12/16.39** | **49.04** |
| ATC w/o Assign | 79.49 | **92.97** | 46.05/49.48 | 21.98/33.03 | 37.17/15.28/33.46 | 24.27/1.81/13.48 | 48.26 |
| ATC w/o Contrast | 79.39 | 92.71 | **48.38/50.78** | 20.37/29.87 | 37.31/15.39/33.65 | 26.40/2.33/15.42 | 48.38 |

experiments, the learning rate is set to 5e-4, the optimizer is set to AdamW (Loshchilov & Hutter, 2019) with $\beta_1 = 0.9$, $\beta_2 = 0.999$ and weight decay of 0.01.

We develop the proposed ATC based on FederatedScope (Xie et al., 2023). In the ASSIGN stage, the training round number is set to 200 and each training round contains 50 optimization steps. The batch size is set to 64, and the number of clusters is set to 5. In the CONTRAST stage, the training round number is set to 100 and each training round contains 200 optimization steps. The batch size is set to 32, the number of top-$K$ clustered clients is tuned in $[4, 8, 16]$, and the temperature value $\tau$ in equation 3 is set to 1.0. All experiments are conducted on GeForce RTX 2080Ti GPUs.

## 4.4 EXPERIMENTAL RESULTS

**Comparisons** The empirical comparisons between the proposed ATC and several baseline methods are demonstrated in Table 2. The performances of FEDAVG-ST outperform those of FEDAVG on most of the adopted datasets, which demonstrates that applying vanilla FEDAVG to construct a FL course among clients with heterogeneous learning objectives can lead to sub-optimal performance of the learned global model.

Further, from the comparisons between personalized FL methods and FEDAVG, we can observe that personalized FL methods help alleviate the negative impact caused by the heterogeneity (i.e., the non-IID data distributions and various learning objectives) among clients. However, it is worth noting that all these baseline methods do not consistently improve upon the performance of ISO-LATED for clients with various learning objectives. Such experimental results suggest that not all participants can benefit from shared knowledge, which motivates us to propose the ATC framework.

The experimental results in Table 2 demonstrate that ATC framework outperforms baselines by a noticeable margin. The results show that ATC achieves consistent improvements on all the participants compared to ISOLATED, and gains an overall 3.66% improvement. Additionally, the proposed ATC outperforms baseline methods by noticeable margins on most of the adopted datasets, particularly on question answering (e.g., 0.74% of improvements evaluated by EM on NewsQA compared to the best baseline method) and text generation tasks (e.g., 2.18% improvement evaluated by RL on MSQG compared to the best baseline method). We conduct independent t-test between the results of ATC and the strongest baseline and get $p < 0.01$. These experimental results confirm the effectiveness of ATC in helping heterogeneous clients to participate and benefit from a FL course.

**Ablation Study** In this part, we conduct a comprehensive ablation study to show the contributions of ASSIGN and CONTRAST in the proposed ATC framework. Firstly, we skip the ASSIGN training stage (denoted as "ATC w/o ASSIGN") or replace CONTRAST training stage with vanilla FEDAVG (denoted as "ATC w/o CONTRAST") in ATC to show their effects, and the experimental results are summarized at the bottom of Table 2. From these results, we can observe that the overall performance of "ATC w/o ASSIGN" and "ATC w/o CONTRAST" drops 0.78% and 0.66% compared to ATC, respectively, which confirms their contributions to the proposed framework.

Table 3: Experimental results of ablation study.

| Method | Classification | QA | NLG | Average |
|---|---|---|---|---|
| VANILLA | 85.67 | 33.81 | 18.36 | 45.95 |
| + MLM | 85.74 (+0.07) | 35.31 (+1.50) | 20.97 (+2.61) | 47.34 (+1.39) |
| + DR | 85.88 (+0.21) | 36.71 (+2.90) | 21.85 (+3.49) | 48.15 (+2.20) |
| + MLM & DR | 86.00 (+0.33) | 36.70 (+2.89) | **21.92 (+3.56)** | 48.20 (+2.25) |
| + MLM & DR & Clustering | **86.05 (+0.38)** | **37.35 (+3.54)** | 21.75 (+3.39) | **48.38 (+2.43)** |

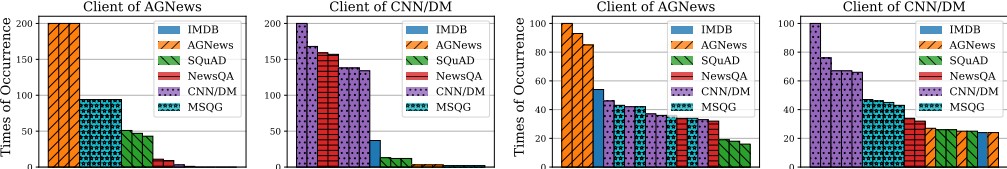

Figure 4: Clustering results in ASSIGN.    Figure 5: Aggregation results in CONTRAST.

Further, we perform a quantitative comparison to analyze the effectiveness of the adopted pre-training tasks, i.e., Masked Language Modeling (MLM) and Denoising Reconstruction (DR), and the clustering algorithms (Clustering) in the ASSIGN training stage. Specifically, we firstly replace the CONTRAST training stage with the vanilla FEDAVG to remove the effect of CONTRAST and skip the ASSIGN training stage in the proposed ATC, which is denoted as VANILLA. Then we gradually add the assigned tasks and clustering algorithms to recover the ASSIGN training stage.

The experimental results are shown in Table 3, from which we can observe that both MLM and DR have positive effects on the performance of ATC. For example, the overall performance of VANILLA has been incrementally improved by 1.39%, 2.20%, and 2.25% by using MLM, DR, and both of them, respectively. Besides, applying the clustering algorithms gains a further 0.18% improvement over "+ MLM & DR" and 2.43% improvement over VANILLA.

## 4.5 FURTHER DISCUSSIONS

**Federated Aggregation in the ASSIGN Training Stage**   We record the statistics of clustering results and illustrate in Figure 4 to show the aggregation results in the ASSIGN training stage. From the results, it is not surprising to observe that clients of the same datasets are mostly clustered together. For example, the clients of AGNews are mostly aggregated with those clients of AGNews. Additionally, clients with similar domain corpus are more likely to be aggregated together than those with different domain corpus. For example, clients of AGNews are more likely to be aggregated with clients of MSQG and SQuAD because they are all collected from the news domain and have similar vocabulary distribution. These results indicate that clients learn useful knowledge from others with similar data/vocabulary distribution when using clustering aggregation in the ASSIGN training stage.

**Federated Aggregation in the CONTRAST Training Stage**   We illustrate the aggregation results (i.e., the $K$ closest neighbors) in Figure 5. From the figure, we observe that clients of the same datasets are most likely to be aggregated with each other, which is similar to the observations in ASSIGN. Besides, clients with similar learning objectives are also aggregated with high probability in the CONTRAST training stage. For example, it can be observed that clients of CNN/DM are aggregated with clients of MSQG since both CNN/DM and MSQG are text-generation tasks. Such phenomena can be interpreted by the fact that clients who perform local training with similar learning objectives generate similar model updates (i.e., consistent gradient directions), therefore they become the $K$ closest neighbors to each other. These results further confirm that clients can learn useful knowledge from other clients whose generated model updates are similar in the CONTRAST training stage of the proposed ATC.

**The Effect of Contrastive Learning Objective**   In this part, we further demonstrate the effect of contrastive loss (CL) defined in  equation 3 in helping clients with similar learning objectives to be

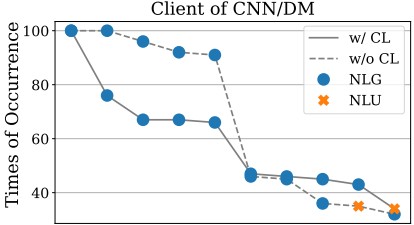 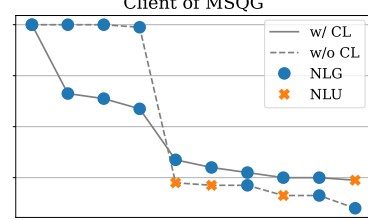

Figure 6: Comparisons between applying and not applying contrastive loss in CONTRAST.

aggregated and learn useful knowledge from each other in the CONTRAST training stage. Specifically, in Figure 6, we plot the neighbors of clients of CNN/DM and MSQG (both are NLG tasks), ranked by the distances measured by the cosine similarity of their contributed model updates. By comparing "w/ CL" with "w/o CL", we can observe that when applying the contrastive loss, clients with similar NLG tasks obtain a high ranking more frequently than that of "w/o CL". Furthermore, clients with NLU tasks hardly rank in the top 10 when the contrastive loss is applied. Since a higher rank implies a higher probability of being aggregated, these experimental results demonstrate that applying contrastive loss can enhance the aggregation among clients with similar learning objectives while suppressing the aggregation among clients with very different tasks.

## 5 RELATED WORKS

Federated Learning (FL) (Konečnỳ et al., 2016; McMahan et al., 2017) has become one of the most popular topics in both research and industrial communities in recent years. In the Natural Language Processing (NLP) community, FL has been applied to various practical scenarios (Hard et al., 2018; Ge et al., 2020; Qin et al., 2021; Chen et al., 2021; Passban et al., 2022). Recent works on FL for NLP can be broadly categorized into the following main aspects: (i) Preserving data privacy. For example, Sui et al. (2020) proposes a federated learning framework for medical relation extraction to protect patient's private information. Qin et al. (2021) incorporates topic memory into the FL framework to overcome data isolation limitations. (ii) Addressing data heterogeneity. For example, Chen et al. (2021) proposes a personalized backbone-patch architecture to address the non-IIDness of question-answering data. Passban et al. (2022) presents a dynamic pulling FL method to efficiently train mixed-domain translation models. We focus on enabling participants with heterogeneous NLP tasks via FL and propose a novel framework ATC in this study.

Towards handling various heterogeneity of clients in federated learning, personalized federated learning (T. Dinh et al., 2020; Fallah et al., 2020; Chen et al., 2022) has been widely studied to meet the demands of real-world applications, such as the non-IID distributions among clients' local data, different system resources of participants, and so on. The existing personalization techniques in FL including regularization (T. Dinh et al., 2020; Li et al., 2020; 2021a), model mixture (Mansour et al., 2020; Deng et al., 2020; Li et al., 2021b), meta-learning (Khodak et al., 2019; Fallah et al., 2020), transfer learning (He et al., 2020; Zhang et al., 2021), clustered learning (Ghosh et al., 2020; Sattler et al., 2021), multi-task learning (Smith et al., 2017; Marfoq et al., 2021; Xie et al., 2022). The proposed framework encourages clients with heterogeneous NLP tasks to exchange useful knowledge learned from similar domain data or downstream tasks.

## 6 CONCLUSIONS

In this paper, we propose a novel FL framework named ASSIGN-THEN-CONTRAST (denoted as ATC) to help participants with heterogeneous NLP tasks learn useful knowledge from each other. The proposed framework consists of an ASSIGN training stage and a CONTRAST training stage. The proposed ATC framework demonstrates a notable improvement (of at least 1.5%) in the overall performance across six diverse NLP tasks. By proposing ATC, we aim to further promote the usage of FL in real-world NLP applications and inspire the community to develop new algorithms for coordinating heterogeneous participants.

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
