# OpenReview forum: "Collaborating Heterogeneous Natural Language Processing Tasks via Federated Learning"
_ICLR.cc/2024/Conference — Submitted to ICLR 2024_

### Official Review · Reviewer_4Nhg · 2023-10-31

**Soundness:** 3 good
**Presentation:** 3 good
**Contribution:** 3 good
**Rating:** 6
**Confidence:** 2

**Summary:**

This paper seeks to introduce a framework, which allows for federated learning to take place between end-users who have different learning objectives (e.g. classification versus generation). They propose a 2 step framework, which should protect the privacy of the server (where all of the learning is aggregated as a part of federation learning), as well as the end-users, at the same time as improving performance across varying tasks.

**Strengths:**

*** Disclaimer *** This paper is very much outside my own specialty, so I've had some trouble understanding the underlying motivations and practical use-cases. That said, my initial review will need to be considered with this context:

Despite being on a topic outside of my wheelhouse, I found this paper mostly quite clear. The authors break down and describe in detail the two steps of their federated learning approach (the "Assign" and "Contrast" components), before conducting baseline experiments, to compare their methods with others in this sub-field. I particularly appreciated the simplicity of experiments (e.g. using simple BERT_tiny models, with simple objectives, on well known datasets, etc., as described in S3.1 and S4.1), so that as an outsider to this sub-field, I could rather focus on the paper's innovation ("Assign then Contrast").

For the aforementioned innovation -- I really have a hard time understanding the impact of this new approach, within the sub-field. If most federated learning can indeed only work on homogeneous tasks, then it seems like quite an accomplishment indeed! And although I didn't understand everything, the explanation is thorough enough that it seems this approach could be loosely implemented by others, following the details of the paper.

Also in the end, I appreciate the discussions, with concrete take-aways and reflections from the analysis, so that I can better understand how something like this could be useful.

**Weaknesses:**

As an outsider to this specific sub-field it is very hard for me to asses the true weaknesses of the paper. Here, I will attempt to make my best guess:

While the "Contrast" part of the proposed method is focused around preserving privacy, it would have been nice to have some verification of this, as privacy is absolutely central to the proposed model. I genuinely do not work in privacy+fairness, so I do not know if leveraging contrastive learning is sufficient for protecting privacy.

One last thing -- which is NOT a weakness, but I think addressing this point would be nice:

In the Introduction, the paper introduces the need for improved federated learning by essentially complaining that increased data protections hurt "data owners". This doesn't sound great... I for one am very grateful to have protections on my data, that keep companies from using my data in risky ways! Thus, I would encourage a slight re-framing of the introduction, that doesn't frame end-users rights for data privacy as a burden.  The rest of the paper is much better, in terms of framing for privacy protection, though.

**Questions:**

Neat project! I only have a few questions for the authors:

1. Why do you not experiment with more clients? Its unclear to me if these are simulated "clients", or real-life clients. I ask because, for some of the experiments in Table 2, the ATC results are quite close to other tasks, so its unclear to me whether minor improvements are statistically significant with such a small number of clients. Or, perhaps, is a small number of clients intentional?

2. Verification: in real life, you would never use ATC w/o "Contrast", right? Wouldn't this put customer data at risk? Regarding the point I made in the "Weaknesses" section, is contrastive learning alone enough to safe guard against privacy leakage?

I am interested in the answers to these questions, but as far as my final score from the manuscript, I am most likely to reference the other reviews and your rebuttals to them, when making my final decision, again because unfortunately this topic is far from my own research topic in NLP.

---

> ### Author Response · Authors · 2023-11-23
> **Response to Reviewer**
>
> Thank you very much for your appreciation and detailed comments! We make the following responses point by point to address your comments.
>
> > W1: "While the "Contrast" part of the proposed method is focused around preserving privacy, it would have been nice to have some verification of this, as privacy is absolutely central to the proposed model."
> Q2: " Verification: in real life, you would never use ATC w/o "Contrast", right? Wouldn't this put customer data at risk? Regarding the point I made in the "Weaknesses" section, is contrastive learning alone enough to safe guard against privacy leakage?"
>
>
> Thank you for the comments!
> - The main purpose of the "Contrast" part is to group clients based on their local objectives rather than preserving privacy. It is used to encourage clients to exchange useful downstream task knowledge by federated aggregation.
> - For privacy protection, the proposed ATC provides fundamental privacy protection (no sharing of data directly) with the help of the federated learning framework. Besides, privacy protection can be enhanced by privacy protection algorithms (such as differential privacy, homomorphic encryption, etc.) to satisfy flexible protection requirements from different applications. Users can balance utility-privacy-cost according. These privacy protection algorithms can be regarded as plugins and are orthogonal to the proposed framework, thus in this submission, we pay more attention to describing how can collaborate clients with different learning objectives.
>
> We have made the above discussions more clear in the revised paper, thank you again!
>
> > W2: "In the Introduction, the paper introduces the need for improved federated learning by essentially complaining that increased data protections hurt "data owners". This doesn't sound great...."
>
> We sincerely appreciate your valuable suggestions!
>
> We absolutely agree that user private data should be protected and some actions should be taken to keep companies from using personal data in risky ways! Our submission explores a practical solution for training models in a privacy-preserving manner.
> We have modified the introduction according to your suggestions to make it precise and friendly. Thank you very much!
>
> > Q1: "Why do you not experiment with more clients? It's unclear to me if these are simulated "clients", or real-life clients."**
> Thanks for your comment!
>
> - We adopt simulated clients in the experiments.
> - We conduct an experiment with more clients: the number of clients increased fivefold in each dataset. The results demonstrate that the proposed method achieves noticeable improvement by 4.94% and 4.75% on the overall performance compared to PerCFL and SPFL (two of the strongest baseline methods), respectively. These results further confirm the effectiveness of the proposed method.
>
> ---
>
> Thank you again for the detailed comments and valuable suggestions! We have made these modifications in the revised paper and believe this submission has been further improved according to your helpful suggestions, and hope that these responses can address all your concerns and convince you to lean more toward acceptance of the paper.

---

> > ### Comment · Reviewer_4Nhg · 2023-12-04
> > **Response to author's rebuttal**
> >
> > Acknowledged.
> >
> > By reading the other reviewer's comments, as well as your rebuttals, I believe the scores that I have chosen are appropriate.

---

### Official Review · Reviewer_3Hkg · 2023-11-01

**Soundness:** 3 good
**Presentation:** 3 good
**Contribution:** 3 good
**Rating:** 8
**Confidence:** 3

**Summary:**

The paper presents an application of Federated Learning in NLP called Assign-then-Constrast (ATC). This framework facilitates multiple clients (or models) with heterogeneous NLP tasks to not only learn from their own data but also from knowledge of other clients without actual sharing of data amongst different clients.
The federated learning is achieved through two stages:

Stage 1 - Assign: Involves local training with unified objective assigned to each of the client by the server

Stage 2 - Contrast: Clients train with different local learning objective and exchange knowledge with other clients via an additional contrastive learning loss.

The paper reports evaluation on six datasets and highlights the efficacy of using such a paradigm to train multiple clients. The comparison is against competitive baselines. Additionally, the qualitative analysis and ablation studies show the importance and significance of each of the different stages and modules.

**Strengths:**

1. Easy to understand paper with comparison against competitive baselines

2. Ablation study highlights the importance of each of the component being added.

3. State-of-the-art results on multiple tasks learning from shared knowledge by multiple clients.

**Weaknesses:**

1. Missing details on some important sections. See Questions.

2. No statistical significance testing conducted for the results that appear quite incremental compared to the baselines.

**Questions:**

1. In the introduction, kindly provide a concrete example/use case for federated learning with heterogeneous tasks.

2. What does consistent model update mean as given in the abstract?

3. Section 3.1: Shed Lighted -> Based on such insights.

4. Section 3.2 - Extension with Clustering Algorithms: The gradients change with each iteration, so can the clusters change as well, or is it fixed based on the first iteration?

5. Section 3.2 - Extension with Clustering Algorithms - Kindly extend upon the clustering algorithm - provide more details.

6. Section 3.3 - 4th paragraph - “To further..” is confusing. Kindly provide concrete examples to highlight this important stage.

7. Section 3.3 - Mention that the contrastive learning loss is one of the additional objective other than the client - task specific objective.

8. Section 4.2: In FEDAVG what equations govern the federated aggregation? In FEDPROX what is the proximal term

9. Table 2: Are these numbers statistically significant? Kindly conduct a significance testing.

10. In the related work section, please highlight the importance of ATC for each of the different previous works rather than clubbing them in the end.

11. In the conclusion section, kindly mention quantitative results, instead of writing the abstract again.

---

> ### Author Response · Authors · 2023-11-23
> **Response to Reviewer (1/2)**
>
> Thank you very much for your appreciation and detailed comments! We make the following responses point by point to address your comments.
>
> > W: "No statistical significance testing conducted for the results that appear quite incremental compared to the baselines."
>
> Thank you for the suggestions! We have added the statistical significance testing in the revised paper in Section 4.4. Specifically, we conduct an independent t-test between the results of the proposed method and PerCFL (the strongest baseline) to show the significance and get $p< 0.01$.
>
> > Q1: "In the introduction, kindly provide a concrete example/use case for federated learning with heterogeneous tasks."
>
> Thank you very much for the suggestion! Federated learning with heterogeneous tasks has many practical use cases. For example, multiple companies may collect text data from similar domains, such as news, but utilize these data for different downstream tasks, such as event discovery, sentiment analysis, and so on. With the help of federated learning and the proposed framework, these companies can collaboratively train a global model to learn valuable and generalized knowledge from text data.
>
> We have added the example in the revised paper, thank you again!
>
>
> > Q2: "What does consistent model update mean as given in the abstract? "
>
> Thanks for your comment! The term "consistent" means that the update directions are similar and not conflict.
>
> > Q3: "Section 3.1: Shed Lighted -> Based on such insights"
>
> Thanks for your suggestion! We have modified it accordingly.
>
> > Q4: "Section 3.2 - Extension with Clustering Algorithms: The gradients change with each iteration, so can the clusters change as well, or is it fixed based on the first iteration? "
>
> Thanks for your comment! The clusters might change as the gradients change with each iteration.
>
> > Q5: "Section 3.2 - Extension with Clustering Algorithms - Kindly extend upon the clustering algorithm - provide more details."
>
> Thank you for the comments. The clustering algorithm used in this study is an agglomerative clustering algorithm proposed by [1], which serves as an example to demonstrate the extensibility of our framework. In fact, more advanced clustering algorithms can be adopted in the assign training stage. We kindly encourage readers to concentrate on the overall architecture of the framework and refer to the references for further details if necessary. Thank you again!
>
> > Q6: "Section 3.3 - 4th paragraph - “To further..” is confusing. Kindly provide concrete examples to highlight this important stage. "
>
> Thank you for the comments! In section 3.3 - 4th paragraph, we introduce a synthetic data generation approach for contrastive learning. This approach is particularly valuable in scenarios where the proposed framework is applied to downstream tasks involving sensitive data (e.g., healthcare) or non-IID data. It is because finding an eligible public dataset can be intractable.
>
> We have made them more clear accordingly in the revised paper, thanks again!
>
> > Q7: "Section 3.3 - Mention that the contrastive learning loss is one of the additional objective other than the client - task specific objective. "
>
> Thank you for the suggestions! We have modified it accordingly in Section 3.3 for a better understanding of the contrastive learning loss.
>
> > Q8: "Section 4.2: In FEDAVG what equations govern the federated aggregation? In FEDPROX what is the proximal term "
>
> Thank you for the comments!
>
> - In FEDAVG, the equations of federated aggregation can be given as $\boldsymbol{w}^{(t+1)} \leftarrow \boldsymbol{w}^{(t)} + \sum _{i \in [N]} \frac{|D _i|}{\sum _{j\in [N]}|D _j|} \Delta \boldsymbol{w}^{(t)} _{i}$
> - In FEDPROX, the proximal term can be given as $\frac{\mu}{2}||\boldsymbol{w} - \boldsymbol{w}^{(t)}||^2$ (please refer to the algorithm 2 in [2] for more details).
>
> > Q9: " Table 2: Are these numbers statistically significant? Kindly conduct a significance testing."
>
> Thank you for the suggestions! We have added the statistical significance testing in the revised paper in Section 4.4.

---

> ### Author Response · Authors · 2023-11-23
> **Response to Reviewer (2/2)**
>
> > Q10: "In the related work section, please highlight the importance of ATC for each of the different previous works rather than clubbing them in the end."
>
> Thank you for the suggestions! We have modified the related work section accordingly for better highlight the importance of ATC.
>
> > Q11: "In the conclusion section, kindly mention quantitative results, instead of writing the abstract again."
>
> Thank you for the suggestions! We have provided more quantitative results in the conclusion section.
>
> ---
>
> Thank you again for the detailed comments and valuable suggestions! We have made these modifications in the revised paper and believe this submission has been further improved according to your helpful suggestions, and hope that these responses can address all your concerns and convince you to lean more toward acceptance of the paper.
>
> Refs:
> [1] Modern hierarchical, agglomerative clustering algorithms. 2011.
> [2] Federated optimization in heterogeneous networks. MLSys, 2020.

---

> > ### Comment · Reviewer_3Hkg · 2023-11-28
> > **Thanks for updating the relevant sections in the paper**
> >
> > I have read the responses of the queries of all the reviewers and will keep my scores unchanged.

---

### Official Review · Reviewer_6MMF · 2023-11-18

**Soundness:** 1 poor
**Presentation:** 3 good
**Contribution:** 3 good
**Rating:** 5
**Confidence:** 3

**Summary:**

The paper describes a novel framework with heterogeneous/private learning objectives for learning from shared knowledge through federated learning(FL). Participants can build a FL course using the Assign-then-Contrast (ATC) framework without revealing or aligning their own learning objectives. The novelty is that clients can work toward a common goal without sharing their own interests (learning objectives), and by using contrastive learning, they can help clients with similar learning objectives exchange useful information. The server aggregates the trainable parameters of the global model based on the assigned task during each training round. The server aggregates the trainable parameters of the global model based on the assigned task to clients during each training round. When the server assigns tasks to clients, the parameters are updated based on the client's local interest (i.e., when assigned MLM, the encoder is updated and aggregated; and when assigned DR, the updated and aggregated parameters include both the encoder and decoder).

**Strengths:**

+ Framework is relevant to the ICLR community: it achieves improvements on all the participants compared to ISOLATED (overall 3.66% improvement).
+ The experiments are well planned - randomization, splitting, sufficient datasets, different local task
+ Comparison to baselines including vanilla FL SOTA algorithms (FedAvg, FedAvg-ST, FedProx) and personalized FL (DITTO, FedBN, PerlCFL, SPFL), and isolated local training
+ Reproducibly is enabled with the implementation descriptions and details + code availability willingness
+ Overall noticeable margins of improvement on heterogeneous clients to participate and benefit from an FL course.

**Weaknesses:**

+ It is still constrained to the global tasks: depending on the server to assign a global task (meaning that participants need to abide by this global task or has to be intrinsically similar)
+ Some novelty is overclaimed, e.g., FL + multi-task [1] (or heterogeneous tasks in the context of this paper),  there is no comparison with this baseline.
+ Some essential questions and concerns on the baselines:
    + The SOTA of the experiments, e.g., IMDB and perhaps others, seems to be different than your experiments (ISOLATED). did you start with a suboptimal setting, e.g., carefully fine-tuned?
+ There is a concern about whether it is worth the margins of improvement provided by Contrastive Learning in the trade of privacy and communication cost.
    + Additional communication cost between clients means $\Theta (K^2 \cdot D)$, where $K$ is client number and $D$ is data size
    + Privacy can be disclosed to not only the server but also peer clients. Synthetic data is proposed as a way to mitigate the effect, but we want to see more restricted discussion on privacy concerns. Please also consider removing privacy claims if these are not addressed, especially in the abstract and introduction.
    + Willingness of clients' participation: If a global task is imposed by the server and introduced to clients what are the potential drawbacks addressed? i.e. local client training costs (training time and memory cost) and risk (of out of memory) increased.
+ Although there is a marginal improvement it is stated "superior performance". Consider lowering the tone.
+ Many writing issues (see questions).

[1] Smith, V., Chiang, C. K., Sanjabi, M., & Talwalkar, A. S. (2017). Federated multi-task learning. Advances in neural information processing systems, 30.

**Questions:**

How will this framework tackle backdoor attacks?

Non-IID could have more discussion, especially the Non-IID introduced by multiple tasks and datasets.

---
[Writing]:

"an FL" -> "a FL"

"learn useful information": One learns from information. Once it's learned, it becomes knowledge.

"unallowable" ?

FL (acronym): once introduced you should stick to it.

---

> ### Author Response · Authors · 2023-11-23
> **Response to Reviewer (1/2)**
>
> Thank you very much for your appreciation and detailed comments! We make the following responses point by point to address your comments.
>
> > W1: "It is still constrained to the global tasks: depending on the server to assign a global task (meaning that participants need to abide by this global task or has to be intrinsically similar)"
>
> Thank you for the comments! In this study, we have carefully selected global tasks (i.e., self-supervised language model pretraining tasks) with the intention of benefiting clients with heterogeneous NLP tasks, while avoiding any additional potential privacy concerns. These global tasks have been proven to be general enough when focusing on the field of NLP, and would not cause additional constraints.
>
>
> > W2: "Some novelty is overclaimed, e.g., FL + multi-task[1], there is no comparison with this baseline."
>
> Thank you very much for the suggestion regarding the references.
> - We notice that the mentioned paper primarily focuses on addressing issues of high communication cost, stragglers, and fault tolerance in general distributed multi-task learning. However, it may not be effectively applicable to collaboration heterogeneous NLP tasks without careful design, particularly in the case of NLU and NLG tasks. We have accordingly included the references and added the discussions, thanks again!
> - We provide the discussions on the comparison between the suggested FL + multi-task framework and the proposed method in the related work section. Besides, in the experiments section, we provide empirical comparisons between the proposed method with the baseline[2] that adopts a multi-task framework, highlighting the advances of the proposed methods.
>
>
> > W3: "The SOTA of the experiments seems to be different than your experiments (ISOLATED). "
>
> Thank you for the comments! We kindly point out that it is much more challenging under the context of federated learning than centralized training, since each client only owns a part of the entire dataset for training (i.e., ISOLATED). Besides, the adopted model is extremely small considering the communication cost in the simulation.
>
>
> > W4: "There is a concern about whether it is worth the margins of improvement provided by Contrastive Learning in the trade of privacy and communication cost"
>
> Thank you for the comments!
> - In the proposed ATC framework, the contrast training stage is a **critical step that builds upon the assign training stage and complements it**. In the assign training stage, clients perform local training with the assigned self-supervised objectives, and their model updates are more related to their data domains; While in the contrast part, clients train with their own objectives, thus their model updates are more specific to their downstream tasks. It is worth noting that different clients can benefit from the knowledge shared by others who have similar data domains or similar downstream tasks.
> - The proposed ATC provides fundamental privacy protection (no sharing of data directly) with the help of the federated learning framework. Meanwhile, privacy protection can be enhanced by privacy protection algorithms (such as differential privacy, homomorphic encryption, etc.) to satisfy flexible protection requirements from different applications. Users can balance utility-privacy-cost according.
> - The willingness of clients' participation is a really interesting and meaningful topic within the context of FL. We would like to take some exploration of incentive mechanisms in our future work. Thanks again for your valuable suggestions!
>
> We have added the above discussions and made them more clear in the revised paper, thank you again!

---

> ### Author Response · Authors · 2023-11-23
> **Response to Reviewer (2/2)**
>
> > W5: "Although there is a marginal improvement it is stated "superior performance"
>
> Thanks for your comments. The proposed method shows a significant improvement (around 1.5% compared to the strongest baseline) on the overall performance. We have made the description more clear in the revised paper.
>
> > W6: "Many writing issues"
>
> Thank you very much for the suggestions regarding the paper writing. We have addressed these issues accordingly in the revised paper, and also made carefully polished to further improve the writing quality.
>
> > Q1: "How will this framework tackle backdoor attacks?"
>
> Thanks a lot for your comments. The proposed framework can be enhanced by various approaches to tackle backdoor attacks, according to the user's requirements of balancing utility and privacy. On the one hand, inspired by previous study[3], we can utilize some simple yet effective approaches, such as norm clipping and differential privacy, to mitigate the backdoor attacks without hurting the overall performance. On the other hand, robust aggregation rules[4,5] can be applied to filter out malicious model updates.
>
> We have added the above discussions in the revised paper, thank you again!
>
> > Q2: " Non-IID could have more discussion, especially the Non-IID introduced by multiple tasks and datasets."
>
> Thank you very much for your suggestions! The Non-IIDness discussed in the context of FL can be defined as *covariate shift* and *concept shift* [6]: the marginal distributions $\Pr(X)$ or the conditional distribution $\Pr(X|Y)$ can vary across clients. The Non-IIDness becomes more complicated and mixed when it comes to heterogeneous tasks, where clients might have different yet related feature and label spaces.
>
> ---
>
> Thank you again for the detailed comments! We believe this submission has been further improved according to your helpful suggestions, and hope that these responses can address all your concerns and convince you to lean more toward acceptance of the paper.
>
>
> Refs:
> [1] Federated Multi-Task Learning. NeurIPS, 2017.
> [2] Personalized heterogeneous federated learning with gradient similarity. 2022.
> [3] Can You Really Backdoor Federated Learning? NeurIPS 2019.
> [4] Understanding distributed poisoning attack in federated learning. ICPADS, 2019.
> [5] Machine learning with adversaries: Byzantine tolerant gradient descent. NeurIPS, 2017.
> [6] FedBN: Federated Learning on Non-IID Features via Local Batch Normalization. ICLR 2021.

---

> ### Comment · Reviewer_6MMF · 2023-12-04
>
> To w3, you may still try your best effort to align the experiments with feasible baselines.
>
> To w4, adding security protection will exacerbate the cost thus reducing the applicability of this proposal.
>
> To w5, the baseline uses a reproduced score with is lower than the original paper. [Linked to w3]
>
> To Q1, do you have some evidence to support the claim?
>
> Thank you for your responses. I acknowledge they are taken into consideration.

---

### Official Review · Reviewer_q9mK · 2023-11-18

**Soundness:** 2 fair
**Presentation:** 3 good
**Contribution:** 2 fair
**Rating:** 5
**Confidence:** 4

**Summary:**

This paper introduces a new technique for applying Federated Learning on sequence modeling tasks (specifically focusing on NLP applications), for coordinating participants with heterogeneous or private learning objectives. The paper proposes the assign and contrast framework, where it relies on a self-supervised learning task for local learning in the assign stage and the use of contrastive learning in the weights averaging stage. The paper introduces a novel framework, and does extensive evaluation on 6 NLP datesets to show the efficacy of the approach.

**Strengths:**

* The paper does a good job in explaining the related work in the FL space, and motivates a need for their framework to combine heterogenous tasks as opposed to past works with more rigid constraints associated with needing to operate on similar or the same tasks.

* The assign part of the framework was introduced well, especially by allowing local clients to do self-supervised learning based on labels they already had access to but did not need to share with the global server

* The paper does extensive evaluation and ablation studies to demonstrate the strength of the framework introduced

**Weaknesses:**

* The contrast part of the framework lacked rigorous motivation. There was no discussion on the pros and cons of treating the cosine similarity of the model updates as a good measure of grouping the data domains. Are the model updates representative of not only the variety of the data domains but also the heterogeneity of the downstream tasks that are applied on that data? Extensive clustering analysis of model updates would need to be made to motivate this objective. Figure 6 is a step in the right direction, but more rigorous analysis would be useful.

* Looking at table 2 showed the proposed framework in the top performing range among the competitors, but the improvement in many cases did not seem significant compared to the state of the art. For eg, the Squad and MSQG task top performer had results just 2-3 points below the proposed framework. For all other tasks, the results of all top performers were not significantly different.

**Questions:**

* It would be useful to explicitly mention the use of self-supervised learning objectives for the assign portion at the bottom of the introduction section. This would clear up early confusion on how the tasks were assigned without having labels for the local data. Similarly, it would be useful to briefly mention the use of clustering approaches while contrasting.

---

> ### Author Response · Authors · 2023-11-23
> **Response to Reviewer**
>
> Thank you very much for your appreciation and detailed comments! We make the following responses point by point to address your comments.
>
> > W1: "The contrast part of the framework lacked rigorous motivation. There was no discussion on the pros and cons of treating the cosine similarity of the model updates as a good measure of grouping the data domains. Are the model updates representative of not only the variety of the data domains but also the heterogeneity of the downstream tasks that are applied on that data? "
>
> Thank you for the suggestions! The main purpose of the contrast part of the framework is to **group clients based on their local objectives**, encouraging them to exchange useful downstream task knowledge by federated aggregation. For more details:
> - The contrast part is a critical step that builds upon the assign part and complements it. In the assign part, clients perform local training with the assigned self-supervised objectives, and their model updates are more related to their data domains; While in the contrast part, clients train with their own objectives, thus their model updates are more specific to their downstream tasks.
> - We conduct experiments and have further discussions on how we group the clients in these two different training parts, as shown in Section 4.5. The results show that in the assign stage, clients with similar data domains are more likely to be aggregated together. On the other hand, in the contrast part, clients with similar learning objectives are grouped. These experimental results further confirm the main idea behind our design.
>
> We have added the above discussions and made them more clear in the revised paper, thank you again!
>
> > W2: " Looking at table 2 showed the proposed framework in the top performing range among the competitors, but the improvement in many cases did not seem significant compared to the state of the art."
>
> Thanks a lot for your comment. This study focuses on the collaboration of heterogeneous NLP Tasks, making it crucial for the proposed method to **achieve better overall performance across multiple tasks**, rather than just being competitive in one or a few tasks. As demonstrated in Table 2, the proposed method shows a significant improvement (around 1.5% compared to the strongest baseline) on the overall performance. For more details:
> - While some baseline methods may achieve comparable performance in one or a few tasks, they fail in other tasks and consequently obtain lower overall scores, as shown in Table 2. For example, PerCFL (the strongest baseline) performs similarly to our proposed method on AGNews/NewsQA, but causes a significant performance drop of approximately 5%/2% on SQuAD/IMDB.
> - In comparison to FedAvg-ST (which constructs multiple FL courses using FedAvg independently, with each FL course only involving clients with the same or similar tasks), the proposed method consistently achieves improvement. These results demonstrate the effectiveness of our method in collaborating heterogeneous NLP tasks.
>
> > Q: " It would be useful to explicitly mention the use of self-supervised learning objectives for the assign portion at the bottom of the introduction section ... Similarly, it would be useful to briefly mention the use of clustering approaches while contrasting."
>
> Thanks a lot for your suggestions!
> - The self-supervised learning objectives for the assign portion used in this study are Masked Language Modeling (MLM) and Denoising Reconstruction (DR), which have been widely adopted in the field of NLP (please refer to Section 3.2 at the bottom of page 3 for more details). Besides these two objectives, more advanced and general self-supervised learning objectives are also compatible with the proposed ATC framework.
> - The clustering approaches while contrasting can be briefly described as: each client is aggregated with its K closet neighbors based on the gradient similarity. For more details, please refer to Section 3.3, located on page 5 near Equation (3) & (4).
>
> We have added the above description according to your suggestions in the revised paper, thank you again!
>
> ---
>
> Thank you again for the detailed comments! We believe this submission has been further improved according to your helpful suggestions, and hope that these responses can address all your concerns and convince you to lean more toward acceptance of the paper.

---

### Meta-Review · Area_Chair_YFYb · 2023-12-18

**Metareview:**

This paper introduces the Assign-Then-Contrast (ATC) framework to expand the application of FL in NLP. ATC enables clients with different NLP tasks to collaborate and learn from each other. During the Assign training stage, clients undergo local training, and in the Contrast training stage, they train with diverse objectives, exchanging knowledge. Extensive experiments on six datasets demonstrate improvements over baseline methods. However, the initial version of the work raised numerous concerns (although the authors addressed many of them, several persist in the revised version). These concerns include the lack of significance tests (now rectified), questions about whether the contrastive part is necessary (the authors have added more discussions on this aspect), concerns about writing/clarity, as well as privacy issues and communication costs raised by reviewer 2 (additional discussions have been added). Reviewer 3 also highlighted various missing details. Overall, I believe the paper has undergone extensive and significant revisions. Although the authors argue they have fixed most of the issues, I believe the work can still benefit from another round of revisions to make it stronger. For example, taking the time to make more thorough and careful comparisons with possible prior baselines (w3, with some efforts to make the experiments comparable), as suggested by reviewer 2, is a good and very reasonable suggestion.

**Justification For Why Not Higher Score:**

I have reviewed the comments and believe that the work could benefit from another round of evaluation (with more qualified reviewers). The second reviewer, whom I know, has experience in the field and is a part of a team that focuses on FL. However, the reviewer who gave it a score of 8 does not seem to be an expert in this domain. The last reviewer also acknowledged having limited knowledge on this topic. The authors did not respond well to the comments, and I also noticed several writing issues even in their responses.

**Justification For Why Not Lower Score:**

N/A

---

### Decision · Program_Chairs · 2024-01-16

Reject